# A Cross-Sectional Study of Potential Antimicrobial Resistance and Ecology in Gastrointestinal and Oral Microbial Communities of Young Normoweight Pakistani Individuals

**DOI:** 10.3390/microorganisms11020279

**Published:** 2023-01-20

**Authors:** Maria Batool, Ciara Keating, Sundus Javed, Arshan Nasir, Muhammad Muddassar, Umer Zeeshan Ijaz

**Affiliations:** 1Department of Biosciences, COMSATS University, Islamabad 45550, Pakistan; 2James Watt School of Engineering, University of Glasgow, Glasgow G12 8QQ, UK; 3School of Biodiversity, One Health, and Veterinary Medicine, College of Medical, Veterinary and Life Sciences, University of Glasgow, Glasgow G61 1QH, UK; 4Department of Molecular and Clinical Cancer Medicine, University of Liverpool, Liverpool L69 3GE, UK; 5College of Science and Engineering, University of Galway, Galway H91 TK33, Ireland

**Keywords:** antimicrobial resistance (AMR), gut microbiome, ecological assembly, niche breadth

## Abstract

Antimicrobial resistance (AMR) is a major global public health concern mainly affecting low- and middle-income countries (LMICs) due to lack of awareness, inadequate healthcare and sanitation infrastructure, and other environmental factors. In this study, we aimed to link microbial assembly and covariates (body mass index, smoking, and use of antibiotics) to gut microbiome structure and correlate the predictive antimicrobial gene prevalence (piARG) using PICRUSt2. We examined the gastrointestinal and oral microbial profiles of healthy adults in Pakistan through 16S rRNA gene sequencing with a focus on different ethnicities, antibiotic usage, drinking water type, smoking, and other demographic measures. We then utilised a suite of innovative statistical tools, driven by numerical ecology and machine learning, to address the above aims. We observed that drinking tap water was the main contributor to increased potential AMR signatures in the Pakistani cohort compared to other factors considered. Microbial niche breadth analysis highlighted an aberrant gut microbial signature of smokers with increased age. Moreover, covariates such as smoking and age impact the human microbial community structure in this Pakistani cohort.

## 1. Introduction

The discovery of antibiotics was one of the greatest scientific achievements of the 20th century. Many deadly infectious diseases such as typhoid fever, pneumonia, syphilis, and tuberculosis can now be treated using standard antibiotics, improving patient treatment outcomes [1]. Bacteria, however, use many mechanisms to evade antibiotics, giving rise to antimicrobial resistance (AMR) [2]. Overuse of antibiotics, especially in agriculture, livestock, and public health, has fuelled AMR evolution in bacterial pathogens, particularly in Gram-negative bacteria [3]. AMR has now emerged as one of the top 10 global public health threats of the 21st century. In 2019, approximately 4.95 million deaths were estimated to be associated with bacterial AMR [4]. A review by the UK Government proposed that by 2050 AMR could kill 10 million people per year worldwide [5]. Quantifying the global burden of AMR to formulate suitable public health measures is difficult due to the unavailability of high-quality population-based data [6]. In 2015, the World Health Organization (WHO) launched the global antimicrobial resistance and use surveillance system (GLASS) program for AMR surveillance at the global level [7]. The direct and indirect impacts of AMR are expected to mostly fall on low- and middle-income countries (LMICs) due to the highly infectious disease burden and associated mortality [8]. Moreover, they lag behind in active surveillance systems to monitor and control antibiotic administration to the general public [9]. Other key contributing factors include the availability of antibiotics without a prescription (leading to misuse), lack of knowledge, and uncontrolled use of antibiotics in agriculture and livestock [10]. Pakistan is ranked 5th in the world population, with >220 million people, and has a high prevalence of antibiotic-resistant bacteria, which poses a significant regional and global threat [11]. The overconsumption and inappropriate usage of antibiotics have been reported by several studies [11,12,13]. Cephalosporins, tetracycline, macrolides, and quinolones are the most prescribed drug groups in Pakistan [14]. Cephalosporins (3rd generation broad-spectrum antibiotics) are prescribed to 67% of patients in Pakistani hospitals [12]. A recent situational analysis of AMR rates amongst GLASS-specified pathogens in Pakistan reported greater than 50% resistance rates for 3rd generation broad-spectrum antibiotics including cephalosporins and fluoroquinolones among *Klebsiella pneumoniae* and *Escherichia coli* [15]. Moreover, a 2020 study also reported that the clinical isolates of *Acinetobacter baumannii* showed 100% resistance to almost every antibiotic tested, suggesting the extensive occurrence of multidrug-resistant strains of *A. baumannii* in Pakistan [16].

The human gut is a reservoir for various microbes harbouring antibiotic resistance genes, which under certain conditions, can cause infectious disease [17]. In general, the gut microbiota plays a key role in human health, aiding digestion, immunity, nutrient acquisition, and protection against pathogens [18,19]. Exposure to antibiotics can cause dysbiosis (i.e., alteration in the taxonomic composition and overall microbial diversity) in the resident gut microbiota [20]. Some bacterial species possess intrinsic mechanisms to counteract antibiotics and can become enriched postexposure. On the other hand, gut dysbiosis promotes the horizontal gene transfer (HGT) of resistance-conferring genes, fuels the evolution of drug-resistant pathogens, and the spread of antibiotic resistance [21]. It also causes a reduction in the diversity of known beneficial bacterial phyla, such as *Bacteroidetes* and *Firmicutes*, which dominate the gut microbiota of healthy adults [22]. This is a potential concern as the healthy gut microbiota provides colonization resistance from invading pathogenic bacteria [23].

The gut is a highly diverse ecosystem of microorganisms. In this environment, on the one hand, a large proportion of the resident microbes are highly stable and present from early life [24]. On the other hand, external factors such as diet, age, and disease development can impact microbial composition [25,26]. Moreover, gut microbial communities are additionally driven by seemingly random processes such as mutations, births, deaths, and ecological drift [27]. We can elucidate the contribution of these different factors towards how communities are assembled within the gut environment. This is done by employing null modelling tools where broadly, the goal is to see if the underlying structure is driven by deterministic or stochastic (random) processes [28,29]. The application of these tools to the human gut microbiome is still relatively new [30]. Studies have found, perhaps unsurprisingly, that strong deterministic forces can shape the human gut microbiome. However, dispersal limitation is also highlighted as a key process [31]. Stegen et al. (2018) argued that cross-talk between environmental and human microbiomes is required to understand and ultimately manipulate or modulate microbiomes [30]. One technique in environmental or ecosystem analysis is niche breadth, which refers to the environment or constraints within which species or populations can survive [32]. This concept has recently been applied to the human gut [33]. The authors found that specific specialist taxa occupied a niche within the human gut during postbariatric surgery. It is also known that the human gut microbiome can be highly individual [26]. However, it is unclear whether these differences are unique or related to individual habits.

In our previous work, we performed a characterisation study of the gut and oral microbial communities of healthy individuals in Pakistan [34]. In this present study, we use this data to (i) quantify the potential prevalence of AMR genes using PICRUSt2 [35], and (ii) link covariates such as the source of drinking water, dietary habits, and antibiotic use to the impacts on microbial community structure and niche breadth. We then determine what forces drive microbial community assembly in these microbiomes using ecological assembly tools.

## 2. Materials and Methods

Briefly, 32 participants (17 females and 15 males) donated 64 ‘Gut’ and ‘Oral’ samples. Participants were young healthy adults with an age range (18–40) and BMI (18–25 kg/m^2^) with detailed summary statistics for the cohort given in Appendix A. More information on the experimental design, participant recruitment and screening, and sample collection and sequencing analysis is included in our initial characterization work [34]. The individuals recruited represent the major ethnic groups (These are self-reported ethnicities identified by the participants at the time of sample collection, and were based on place of birth as opposed to where they were currently residing.) from Pakistan including Punjabi, Pathan, Saraiki, Kashmiri, Sindhi, and Balochi, and belong to major geographic regions including Punjab, Khyber Pakhtunkhwa (KPK), Baluchistan, Sindh, Azad Jammu and Kashmir (AJK), and the Federal Capital Islamabad, as shown in Appendix A. Out of the total of 32 individuals, 7 were reported to have consumed antibiotics three months or more before sampling. All study participants provided written informed consent and data analysis was performed anonymously. The study was performed under a Human Subjects Protocol provided by the COMSATS ethics review board (E&I Review Services, IRB Study #13044, 5 October 2013). The details of sampling and DNA extraction protocols are given in [34]. The samples were amplified for the V4 region of the 16S rRNA gene and were sequenced on the NextSeq 500 platform producing 2 × 150 bp paired-end reads.

### 2.1. Bioinformatics

The resulting 16S rRNA gene sequences (9,961,420) for *n* = 60 samples were processed with the open-source bioinformatics pipeline QIIME2 [36]. Note, four samples from the oral samples dropped out due to low read numbers, leading to 28 ‘Oral’ samples. Initially, the paired-end sequences were demultiplexed and quality trimmed using a PHRED quality score of 20. A Deblur algorithm within QIIME2 [37] was then employed to recover the Amplicon Sequence Variants (ASVs), which were then aligned to the reference alignment database SILVA SSU Ref NR release v138 [38]. Within QIIME2, the q2-alignment method, MAFFT [39] was used to create multisequence alignment of ASVs. Afterwards, a mask was applied to remove phylogenetically ambiguous alignments to obtain a rooted phylogenetic tree using FastTree [40] within the q2-phylogeny framework. After the bioinformatics steps, we obtained 2293 ASVs for *n* = 60 samples with summary statistics of reads aligned per sample to these ASVs as follows: (1st Quartile: 58,459, median: 124,589, mean: 127,625; 3rd Quartile: 186,816; max: 282,077). PICRUSt2 algorithm as a QIIME2 plugin [35] was used on the ASVs to predict the functional abundance of microbial communities and further downstream predictive AMR analysis. Although the prediction process is highly dependent on the number of pathways available for the reference genomes, with PICRUSt2, by virtue of a more comprehensive reference database (~20,000 genomes), only 2 out of 2293 ASVs were dropped out, not adhering to weighted nearest sequenced taxon index (NSTI) scores of 2.0 (average branch length that separates each ASV from a reference weighted by the abundance of ASVs), and as a result, this high alignment increases confidence in the prediction quality.

### 2.2. Statistical Analysis

The summary statistics of the final metadata are given in Appendix A. Statistical analysis was performed using R software (R Core Team, 2022) [41]. R scripts used for statistical analysis are available at http://userweb.eng.gla.ac.uk/umer.ijaz/bioinformatics/ecological.html (accessed on 1 January 2023) and some as part of R’s microbiomeSeq package http://www.github.com/umerijaz/microbiomeSeq (accessed on 1 January 2023)**.**

#### 2.2.1. Prefiltering

For statistical analysis, any samples with < 5000 total reads were dropped, and also excluding contaminants based on taxonomy (chloroplast, mitochondria, and ASVs unassigned at the phylum level) as per typical recommendations given at https://docs.qiime2.org/2022.8/tutorials/filtering/ (accessed on 1 January 2023).

#### 2.2.2. Microbial Diversity Analyses

The Vegan package [42] was used to calculate alpha- and beta-diversity measures. For alpha diversity measures, we have used: Fisher’s alpha*—*a parametric index of diversity that assumes the abundance of ASVs following the log series distribution; Pilou evenness, which compares the actual diversity values to the maximum possible diversity value, and is constrained between 0 and 1.0, where lower values will indicate more variation in abundance between different ASVs in the community; rarefied richness—the estimated number of species/features in a rarefied sample (to minimum library size); Shannon entropy—a commonly used index to measure balance within a community; and Simpson index*—*a measure of dominance that weighs towards the abundance of the most common ASVs and is less sensitive to rarer ASVs. Richness, Shannon entropy, and Simpson indices are parameterized versions of Hill numbers with q = 0, q = 1, and q = 2, respectively. For beta diversity, we have used principal coordinate analysis (PCoA) plots of ASVs using two different distance measures in Vegan’s cmdscale() function: Bray-Curtis, which is a distance metric that considers only ASV abundance counts; and weighted unifrac, which is a phylogenetic distance metric combining phylogenetic distance with relative abundances. UniFrac distances were calculated using the phyloseq package [43]. Analysis of variance was performed using Vegan’s adonis() function with distance matrices (Bray-Curtis/weighted UniFrac) against sources of variation. This function is referred to as PERMANOVA and fits linear models to distance matrices and used a permutation test with pseudo-F ratios and gives percentage variability in microbial community explained by a particular covariate as R^2^ value (if significant).

#### 2.2.3. Contribution of Antimicrobial Resistance Genes to Microbial Diversity

In an approach similar to Su et al. (2020), we used predictive functional genes to detect predictive antimicrobial resistance genes (piARGs) to provide a potential antimicrobial resistance status of the ‘Gut’ and ‘Oral’ microbial communities [44]. First, we defined potential antimicrobial resistance (AMR) genes as those found from known *Kyoto Encyclopedia of Genes and Genomes* (KEGG) signatures (https://www.genome.jp/kegg/annotation/br01600.html) (accessed on 1 January 2023), denoted as ‘Signature KOs for potential antimicrobial resistance’ and this gave a total of 90 KEGG orthologs (KOs) related to these piARG signatures (Appendix A). Out of this full list of 90 KOs, 52 were observed in our dataset (as matched to the PICRUSt2 functional data), which we have highlighted in grey. To see how these piARGs vary within a cohort (i.e., gut versus oral samples, organized by antibiotics usage status and smoking status), we calculated how much the 52 detected piARGs contributed to beta diversity patterns observed in the dataset. For this purpose, we employed the Bray-Curtis (BC) dissimilarity as a metric of community dissimilarity, and is defined as $BC_{jk}=\frac{\sum \left|X_{ij}-X_{ik}\right|}{\sum \left(X_{ij}-X_{ik}\right)}$, where BC is the Bray-Curtis dissimilarity between communities j and k and X are the relative abundance of the KEGG orthologs i. Since BC is a scaled summation of abundance differences between two communities, we can easily partition BC dissimilarity between two samples attributable to the piARGs (Appendix A). To obtain the contribution of the piARGs, for two samples we calculated the BC twice, first calculating the summation in the numerator of the BC expression using the subset of KOs (BC_subset) relevant to AMR, and once with all the total KOs (BC_all). There was no change in the denominator. Dividing the BC_subset by BC_all then reports the fraction of beta diversity attributed to piARGs. For this purpose, we employed R’s summary package [45]. To see if the Bray-Curtis contribution was significant between groups, we used Tukey’s Honest Significance Difference (HSD) test from R’s Stats package.

#### 2.2.4. Interaction between the Microbial Communities or Predictive Antimicrobial Resistance Genes (piARGs) with Study Participant Variables

To find the relationship between microbial communities or piARGs and sources of variation in study participants (*age, BMI, smoking status, antibiotics usage status, fresh fruits Status, junk food status, and sources of drinking water*), we have used a generalised linear latent variable model (GLLVM) [46], which extends the basic generalized linear model that regresses the mean abundances of microbes or the piARGs against all sources of variation, even those that are not directly observed, as confounding latent variables. GLLVM extends the basic generalized linear model that regresses the mean abundances $\mu_{ij}$ (for i-th sample and j-th microbe or piARG of individual microbes or piARGs against covariates $x_{i}$ by incorporating latent variables $u_{i}$ as $g\left(\mu_{ij}\right)=\eta_{ij}=\alpha_{i}+\beta_{0j}+x_{i}^{T}\beta_{j}+u_{i}^{T}\theta_{j}$, where $\beta_{j}$ are the microbe or piARG specific coefficients associated with individual covariates. A 95% confidence interval of the coefficients (either positive or negative, and not crossing 0) is then interpreted as an increase or decrease in that covariate and causes an increase or decrease in the abundance of the microbe or piARG, and $\theta_{j}$ are the corresponding coefficients associated with latent variable. $\beta_{0j}$ are microbes or piARG specific intercepts, while $\alpha_{i}$ are optional sample effects which can either be chosen as fixed effects or random effects. To model the distribution of individual microbes and or piARGs, we have used negative binomial distribution with an additional dispersion parameter, and used log() as a link function. Additionally, the approximation to the log-likelihood is done through Laplace approximation (LA) with the final sets of parameters in glvmm() function being family = ‘negative.binomial’, method = “LA”, and starting.val = ‘zero’ that seemed to fit well. Analysis was performed for the top 100 most abundant genera and all the or piARG observed in our datasets. In addition, the factor loadings $\theta_{j}$ store correlations of microbes with the residual covariance matrix $\Sigma =\Gamma \Gamma^{T}$, where $\Gamma =\left[\theta_{1}\ldots \theta_{m}\right]$ for m latent variables. This residual covariance matrix gave a co-occurrence relationship between microbes and or piARGs that is not explained by the observed covariates (shown in the Appendix A).

#### 2.2.5. Differential Taxa

Since much of the data were paired, i.e., the same individuals provided two samples, one for the gut, and one for oral samples, we have used a specialised cluster association test [47] utilising R’s miLineage package (https://tangzheng1.github.io/tanglab/software.html) (accessed on 1 January 2023) with this test referred to as a QCAT-C test using the QCAT_GEE. Cluster () function (with default values) from the package. The test is robust enough to deduce complex correlations that exist among microbes due to paired nature of samples. Additionally, the QCAT-C test is a two-part test where it fits separate models to microbes that are excessively zero, and those that are not, referred to as positive microbes, based on the taxonomic tree to localize the covariate-associated lineages (gut male, oral male, gut female, and oral female). As a result, the differential abundance analysis of microbes gives better estimates and reduces type 1 errors. To visualise the differentially abundant taxa at different taxonomic ranks, we have used total sum scaling followed by a centralized log ratio (TSS + CLR) transformation on the raw abundance values.

#### 2.2.6. Microbial Niche Width

To assess microbial niche width, we have used R’s MicroNiche package [33] with the goal of determining positively/negatively associated microbial species with BMI and age by explicitly incorporating the possible set of environments as a parameter (something that simple correlation analyses exclude). We have run the algorithm for age and BMI separately for all possible combinations of ethnicity × antibiotics usage status, as well as ethnicity × smoking usage status, considering these as possible sets of environments a microbe is exposed to, for four different cohorts, i.e., gut male, oral male, gut female, and oral female. Before implementing this approach, we filtered out genera by applying the limit of quantification (LOQ) method. LOQ fits a log abundance distribution and filters out taxa that fall below a decision threshold (1.65 × standard deviation of the fitted distribution), calculated from the distribution of microbes with 95% certainty that these microbes will fall within a null distribution where the mean microbial abundance is zero. With this prefiltering step performed, we then calculated Hurlbert’s $B_{N}=\frac{1}{\sum_{i=1}\frac{p_{i}^{2}}{r_{i}}}$, where $p_{i}$ is the proportional abundance of a microbe in the i-th environment, and we have an additional $r_{i}$ proportional covariate data (BMI and Age) in the formula. The model yields a value between 0 and 1 for each microbe and corresponding covariate, indicating whether there is an inverse (~0) or a positive relationship (~1), with 0.5 indicating no relationship to the covariate. To obtain *p*-values, a null modelling procedure was considered by generating a random normal distribution of 999 possible Hurlbert’s $B_{N}$, and by tagging it as “negative” if its $B_{N}$ < 5th quantile, and “positive”, if its $B_{N}$ > 95th quantile, as per the original author’s instructions. Finally, to visualise the results, we have used R’s igraph [48] to draw the network of relationships between the microbe and the observed covariates.

#### 2.2.7. Null-Modelling Analyses

To understand the ecological mechanisms that are driving microbial community assembly in the gut and oral communities, we used various null modelling tools. Briefly, we used the nearest taxon index (NTI) to determine whether the community was structured due to strong environmental pressure based on local clustering in the phylogenetic tree. NTI is typically preferred for microbial datasets due to the presence of significant phylogenetic signals across short phylogenetic distances [49]. We further used the normalised stochasticity ratio (NST) to determine the ratio of stochasticity in the communities [28]. Taxa-richness constraints of proportional–proportional (P–P) and proportional–fixed (P–F) were applied for each metric. We have also used the permutational multivariate ANOVA (also referred to as PANOVA) to obtain significance for NST between different cohorts. We then applied quantitative process estimates (QPE) to quantify the contribution of specific assembly processes. This is based on an ecological framework that breaks down assembly processes into variable or homogeneous selection, dispersal limitation or homogenising dispersal and undominated (neither dispersal nor selection processes dominate) [28]. The variable selection gives rise to high compositional differences in community structure due to different environmental/host conditions. In contrast, homogenous selection occurs when these conditions result in consistent pressure. dispersal processes refer to the movement of microbes throughout space, whether they are absent (dispersal limitation) or present with high rates (homogenising dispersal), resulting in homogenisation.

To calculate the nearest taxon index (NTI) we have used R’s picante package to calculate it using mntd() and ses.mntd() functions [50]. We have used 999 randomisations using null.model = “richness” in the ses.mntd() function and have only considered taxa as either present or absent regardless of their relative abundance. Positive values of NTI indicate that species co-occur with more closely related species more frequently than expected by chance, with negative values suggesting otherwise. For NTI, values > +2 indicate strong environmental pressure (determinism), and values < −2 indicate strong competition among species as the driver of community structure. We applied the quantitative process estimates (QPE) framework to quantify the contribution of different ecological assembly processes [28]. This is first assessed by deviation from the observed βMNTD (β-mean nearest taxon distance). The mean of the null distribution was then evaluated using βNTI (β nearest taxon index). When the observed value of βMNTD deviated significantly from the null expectation, the community is assembled by variable (βNTI > +2) or homogenous (βNTI < −2) selection processes. If the difference is not significant, the observed differences in phylogenetic composition are considered to be the result of dispersal mechanisms enabling ecological drift. These are differentiated using the abundance-based β_RC_ and a Bray-Curtis dissimilarity metric for beta diversity (β_RCbray_). Dispersal limitation contributes to the community assembly. The β_RCbray_ > +0.95 homogenising dispersal will be contributing if β_RCbray_ > +0.95 and community turnover is due to undominated mechanisms if β_RCbray_ was between −0.95 and +0.95. 

#### 2.2.8. Complexity Stability Analysis

To understand the complexity–stability relationship of the gut and oral cohort, we have followed the computational framework of Yonatan et al. (2022) for estimating the complexity of microbial ecosystems [51]. Typically, an ecological system with n interacting species can be modelled with a generalised lotka volterra model of the form $\frac{dx_{i}}{dt}=x_{i}\left(r_{i}+\overset{n}{\underset{j=1}{\sum }}A_{ij}x_{j}\right)$, where $x_{i}$is the abundance of species i, $r_{i}$ is the intrinsic growth rate of species i, and $A_{ij}$ is the interaction coefficient, i.e., the effect of species j on species i. Traditionally, this matrix was obtained through network inference approaches that usually do not capture the causal relationships between interacting species in $A_{ij}$. Once the interaction matrix is obtained, the stability–complexity relationship of the ecosystem can be derived as a curve that satisfies $\alpha \sqrt{nC}<1$ (also called May’s stability criteria [52] where $\alpha^{2}$ and C are the variance and density (“connectance”) of the nonzero off-diagonal elements of $A_{ij}$. The effective connectance $D^{2}$ is then inferred (where $D^{2}\propto \alpha^{2}C$) without the need to infer a co-occurrence relationship, which is obtained by the slope of regression fitted to the dissimilarity/overlap plot to the 25% overlap values for the paired-wise dissimilarity/overlap values for N samples in a given category (oral or gut) from a total of $N\left(N-1\right)/2$ paired-wise values. To calculate the dissimilarity between samples, we have used a Spearman correlation, whereas to calculate the effective number of species, we used richness as an exponential of Shannon entropy. All figures in this study were generated using R’s ggplot2 package [53].

## 3. Results

### 3.1. Diversity Patterns Differ across Gender and Sample Types

Generally, species richness (total number of expected species) in the gut samples was higher than in the oral samples, and the gut of female participants showed a higher richness than those of males (Figure 1A). These differences were statistically significant for Pielou’s evenness (balance in species numbers) and Simpson (putting more emphasis on abundant species) indices. When considering beta diversity through abundance counts alone, we can see two distinct clusters of oral and gut samples, with ~28% variance explained by sample types using PERMANOVA (Figure 1B). Whilst there was a marginal difference (*p* = 0.059) in microbial composition between males and females, accounting for a 2% variation, these differences became significant (*p* < 0.001) when phylogeny was considered (weighted UniFrac). This results in two distinct gut microbial profiles from males and females, retaining an overall ~26% variation between the gut and oral samples (Figure 1B).

### 3.2. Intersample Variability in AMR Genes Is Attributed to Both Antibiotic Usage and Gender

Signatures of predictive ARG composition were similar and much lower in contribution than any other cohort for oral samples where antibiotics were not used (Figure 2). However, the contributions were highly variable within sample groups, though we observed less variability in oral samples as compared to gut samples. For example, the contribution of piARGs ranged from 0.05% to 0.3% in the guts of females who declared they had not taken antibiotics, as compared to 0.02% to 0.1% for the female oral samples. In terms of smoking status, a similar trend in piARG composition was found for most sample types, except the female samples which were lower in contribution. However, these results are inconclusive due to the smaller sample size of females who smoked (*n* = 3).

### 3.3. Sources of Variations Implicated in the Prevalence of Antimicrobial Resistance Genes

We then analyzed the abundance of piARGs against all sources of variation using a generalized linear latent variable model (GLLVM) to find the covariates that on average caused a substantial change in the abundance of piARGs. Interestingly, in the participants that declared antibiotic use status “Yes” (7 gut and 6 oral samples), only 5 piARGs (out of 52) were found to have a strong positive association with antibiotic use (Figure 3; K19217, K19322, K19276, K17881, and K19100). These piARGs belong to beta-lactamase and aminoglycoside drug groups and are associated with pathogens that are classified as serious threats by the Centers for Disease Control and Prevention (CDC), including multidrug-resistant *Acinetobacter* and multidrug-resistant *Pseudomonas aeruginosa*. Out of 52 piARGs, 11 were found to be increased with increasing BMI. Most of these piARGs belong to beta-lactamases, including narrow-spectrum penicillin and extended-spectrum cephalosporin drug groups. Drinking water sources with tap water showed 29 positively associated piARGs compared to 3 with bottled mineral water. Most of these piARGs were associated with serious threat pathogens (B6, B7, B1) and some with urgent threats (A2 and A3). Pathogens that were urgent threats included A2: carbapenem-resistant *Enterobacteriaceae* (CRE) and A3: drug-resistant *Neisseria gonorrhoeae*.

In terms of sample types, there were more positive associations of piARGs with gut samples (33/52) as compared to oral samples (7/52). These were primarily from beta-lactamases and aminoglycoside drug groups. Positive beta coefficients for junk food (high carbohydrate and high-fat foods) were shown in 14 piARGs. Individuals consuming fresh fruits showed only five piARGs with positive beta coefficients. Detected piARGs varied with respect to ethnicity. Fewer piARGs with positive beta coefficients were found in individuals from the Pathan group (3/52), as compared to Punjabi individuals (11/52). There was little variation in piARGs between genders, with 5/52 piARGs with a positive association with males and 3/52 piARGs with a positive association with females. Each of these piARGs belonged to beta-lactamases (Figure 3). We also found that many of the piARGs detected were co-occurring (Appendix A).

### 3.4. Key Microbes Associated with Sources of Variation

We again applied GLLVM, in this case, to find the microbial taxa that were strongly associated with the variables that showed strong positive associations with piARGs. The taxa that were positively associated with drinking tap water were *Brachyspira*, *Phascolarctobacterium*, *Butyrivibrio*, *Succinivibrio*, *Gastroanaerophilales*, *Neisseria*, *Strepobacillus*, *Cardiobacetrium*, and *Tannerella* (Figure 4). *Brachyspira* taxa showed a strong positive association with the male gender and, additionally, the use of antibiotics. *Phascolarctobacterium* also showed a strong positive association with antibiotic use. While we observed reduced *Megamonas* abundance with antibiotic use, and a similar negative association with increasing age. In contrast, we observed a strong positive association between *Megamonas* with smoking. *Lachnoanaerbaculum*, *Bergeyella*, *Tannerella*, *Cardiobacterium*, *Stomatobaculum*, *Streptobacillus*, *Pseudopropionibacterium*, and *Selenomonas* taxa groups were all positively associated with oral samples (Figure 4). Additionally, we observed differences in the microbial taxa with ethnicity, with Saraki individuals showing a positive association with *Brachyspira* and other taxa, which were negatively associated with Punjabi, Pathan, and Kashmiri.

Our data also highlighted certain microbial taxa co-occurrences (Appendix A). For example, *Neisseria*, *Actinobacillus,* and *Megasphaera* showed an inverse co-occurrence with other bacterial taxa including *Ruminococcus*, *Lachnospira, Bifidobacterium*, *Blautia*, and *Paraprevotella*. While, *Alistipes* showed a strong positive co-occurrence with *Bifidobacterium*, *Faecalibacterium*, and *Ruminococcus*.

### 3.5. Microbial Niche Exploration Reveals Signature Taxa for Oral and Gut Communities Associated with Age and BMI

To determine how the environment shapes the distribution of taxa (i.e., what niche dictates which taxa should proliferate in a specific environment), we performed microbial niche breadth analysis. This analysis was conducted using the numerical covariates (age and BMI) and was further parameterized by ethnicity and antibiotics usage status or smoking status. In terms of age and BMI, we observed three clusters, one for the gut, and another one for oral samples as well as a third cluster, where genera change for the male gut with increasing age, when the environments are a combination of ethnicity and smoking status (Figure 5). Notably, the latter cluster shows only negative associations with *Megasphaera*, *Brachyspira*, *Phascolarctobacterium*, and *Anaerococcus*, for example, while the former two clusters showed a mix of positive and negative associations with a wide range of taxa (Figure 5). Some of the genera that lie between the clusters for gut and oral samples simultaneously change in both types of samples (inverse or concomitant relationship between gut and oral) with respect to age and BMI: *Treponema*, *Porphyromonas*, *Streptococcus*, *Corynebacterium*, and *Campylobacter* were all inversely related; and *Prevotella* and *Alloprevotella* otherwise. With respect to the third cluster, there are genera from male gut samples that are negatively associated with age (when the environments are parameterized in the model as a combination of ethnicity and smoking status) and are not associated with any other cohort. These include *Finegoldia*, *Elusimicrobium*, *Peptoniphilus*, *Libanicoccus*, *Phascolarctobacterium*, *Anaerococcus*, *Tyzerella*, and *Lachnospiraceae [Ruminococcus] gnavus* group. Some genera were associated positively with BMI for male gut samples (including a combination of ethnicity groups and antibiotic usage status) and were not associated with any other cohort. These include *Romboutsia*, *Mitsuokella*, *Murdochiella*, *Parabacteroides*, *Desulfovibrio*, *Clostridium_sensu_stricto_1*, and *Oscillospirales;[Eubacterium] coprostanoligenes group*. Considering BMI as a property (a continuous covariate observed in all possible environments) for gut male samples, there were a total of six environments observed in terms of ethnicity and antibiotics usage status (Appendix A), with the only microbial genus whose abundance decreased with increasing BMI was *Treponema*.

### 3.6. More Environmental Pressures on Oral Communities than Gut Communities

The above results highlighted that sample type was a distinct niche for specific taxa groups. Therefore, in this next section, we wanted to determine what could be driving the observed patterns in microbial communities. Using the nearest taxon index (NTI), we confirmed that for both gut and oral communities, a strong environmental pressure (NTI > +2) is exerted on these communities that are dependent on the host environment they reside in (Figure 6A). Relatively higher environmental pressure was observed in oral communities than in gut communities. We applied normalized stochasticity ratio (NST) analysis (Figure 6B) to confirm this observation where the oral communities were found to be more deterministic. Between genders, female oral communities are more deterministic than male oral communities with a marginal significance in terms of PANOVA. However, the converse trend was observed for gut communities where, in the guts of females, the most stochastic amongst all the cohorts considered. Dispersal limitation was the largest contributor to microbial assembly (Figure 6B). In terms of deterministic measures of microbial assembly, ~12% of the oral male microbiome was explained by homogeneous selection suggesting similar environmental oral conditions. In contrast, ~10% of the female gut microbiome was explained by variable selection, suggesting the existence of multiple stable gut environments.

### 3.7. Oral Microbiota Is Less Stable as Compared to the Gut Microbiota

Higher effective connectance suggests that any local perturbation in one or a few species within oral microbiota can affect the entire community more substantially, making it less stable when compared with gut microbiota. We have found that the effective connectance D^2^ value was higher for oral samples as compared to gut samples (Figure 7A), suggesting that oral communities are less stable when compared with gut microbial communities. D^2^ value is calculated using the regression coefficients (slope), which for oral communities was 1.83 (Figure 7B) compared to 0.70 for gut communities (Figure 7C).

### 3.8. The Link between Oral and Gut Microbial Communities

Interestingly, in those subjects (paired gut–oral samples from subjects are indicated using connected lines) that show specific taxa as outliers in the gut, these taxa were typically also highlighted as outliers in the subjects’ paired oral sample (e.g., at family level: *Bifidiobacteriaceae*, *Butyricicoccaceae*, *Clostridiaceae*, *Microbacteriaceae*, *Neisseriaceae*, *and Saccharimonadaceae* [Appendix A; and at genus-level; *Alloprevotella*, *Campylobacter*, *Cardiobacterium*, *Bergeyella*, *and Neisseria* [Appendix A). This differential analysis can also be used to determine the differentially abundant taxa between genders. For example, prominent taxa in the gut communities included *Elusimicrobium*, *Phascolarctobacterium and Succinivibrio*, which were in high abundance in males versus females (Appendix A). In contrast, other prominent gut taxa, *Alistipes*, *Bacteroides*, and *Bifidobacterium*, were high in female samples and low in male samples. Similarly, for oral communities, *Peptostreptococcus*, *Streptobacillus*, *Catonella*, *Cardiobacterium*, *and Pseudopropionibacterium* were high in male samples and low in female samples. While *Centipeda*, *Monoglobus*, *Oscillibacter*, *Lachnospiraceae UCG 10*, *Desulfovibrio*, *and Olsenella* were in higher abundance than male oral communities (Appendix A).

## 4. Discussion

Our results highlight that piARGs increased in the gut as compared to the oral samples, which is in agreement with the literature and likely explained by differences in the microbial host and mobile genetic element associations [54]. In contrast, Carr et al. (2020) found that oral human microbiomes contained a higher abundance of ARGs than gut samples [55]. However, our work is also consistent with the previous work highlighting the gut as a reservoir for antimicrobial resistance, although it did not consider oral communities [56]. We used GLLVM to statistically determine the associations of piARGs with key variables. This revealed that more piARGs were associated with an increased BMI and notably were the highest in individuals who drank tap water. Around 80% of the Pakistani population is exposed to unsafe drinking water because of the scarcity of safe drinking water sources or lack of treatment facilities [57].

The piARGs detected were potentially associated with multidrug-resistant pathogens that belong to serious and urgent CDC threat levels. The majority of these were associated with cephalosporin resistance. This group of antibiotics is widely prescribed in Pakistani hospitals (67%) [12]. In addition, fluoroquinolones are also the most prescribed antibiotic group in Pakistan [58]. The presence of piARGs in our study linked with carbapenem-resistant *Enterobacteriaceae*, *Acinetobacter baumannii*, and drug-resistant *Neisseria gonorrhoeae* is also alarming. Bilal et al. (2021) reported that these bacteria show complete resistance towards a wide range of antibiotics in Pakistan including ceftriaxone, ciprofloxacin, and cefepime [11]. We thus speculate that poor treatment of hospital or sewage wastewater or agricultural run-off may be a source of contamination in the drinking water in Pakistan [57,59,60]. In general, in LMICs, the most deadly and common pollutants in drinking water are of biological origin [61]. For example, a study by McInnes et al. (2022) observed human waste as a primary source of AMR in Bangladesh [62]. Future work to include the sampling of groundwater and tap water for antibiotic detection and microbial analysis may provide more insight into drinking water’s role in spreading AMR.

A potential limitation of this work is that the piARG results are based on metabolic profiles predicted through PICRUSt2, which have not been confirmed through quantitative PCR or other measures. Moreover, associations with specific taxa need to be confirmed due to the potential for antibiotic-resistance encoding genes to transfer between bacterial species [63]. Nonetheless, PICRUSt2 has been shown to perform well on human-associated microbiome datasets. This is mainly due to a comprehensive reference database of genomes whose functions are already known (a 10-fold increase in the numbers since the previous release) [35]. Shotgun metagenomics of these samples would more accurately highlight ARG prevalence; however, the experimental cost and resources for processing and data analysis may be prohibitive for LMICs, and predictive modelling may be a viable monitoring option for determining potential AMR spread.

We further linked microbial taxa to individual variables (age, BMI, smoking, antibiotic usage status, ethnic groups, and source of drinking water). Of these, *Megamonas* taxa had the most significant associations, being negatively associated with age, and positively associated with BMI and smoking. These are all in agreement with the literature [64,65,66]. *Megamonas* abundance is linked with high caloric intake and restricted activity [64]. Particularly, it is implicated in smokers and those who have a high BMI, which may likely be associated with endotoxemia and an inflammatory phenotype of the gut which is inherent in such subjects [67]. More importantly, *Megamonas* species are involved in glucose fermentation into propionate and acetate [68], and these short-chain fatty acids (SCFAs) are beneficial for health [69]. Antibiotics alter the balance of SCFAs, and cause dysbiosis in the gut environment [70].

We observed a strong positive association between antibiotics intake and drinking tap water with the *Brachyspira* taxa. There are a number of studies linking *Brachyspira* with diarrhoea and abdominal pain from faecal-oral transmission [71,72]. Studies have proposed that faecal-contaminated water is an important route of *Brachyspira* transmission [73]. Moreover, we identified a strong positive correlation for *Neisseria* genera with drinking tap water. This correlates to piARGs detected in our dataset corresponding to drug-resistant *Neisseria gonorrhoeae*, however, this link cannot be confirmed as amplicon sequencing cannot provide species or strain-level information. Nonetheless, even commensal *Neisseria* species have been linked to AMR transmission [74]. Therefore, our findings merit further investigation of the drinking water microbiome in Pakistan for understanding AMR transmission and gut health.

Using the microbial niche breadth approach, we highlighted that the gut and oral communities were distinct (when separated according to age and BMI). Interestingly, the QCAT results showed that for some taxa, the subjects that were implicated as outliers in the gut were also outliers in the oral samples suggesting that dysbiosis in the gut may also manifest in oral communities, e.g., the literature suggests that the reported genera in our study, including *Alloprevetolla* and *Campylobacter*, are found in both gut and oral microbiota. The presence of *Campylobacter* in the oral microbiome is also linked to initiating inflammatory bowel disease (IBD) [75]. Indeed there are some recent studies where the gut–oral axis is explored [76,77]. We have also found piARGs with positive co-occurrences in both gut and oral communities. This warrants further investigation in a clinical setting as the use of oral communities as a proxy for gut communities and gastrointestinal system diseases would be more convenient [78].

QCAT, GLLVM, and niche breadth analyses highlight that certain bacterial genera, including *Megamonas*, *Phascolarctobacterium*, and *Succinivibrio,* have high abundance in males, while *Bifidobacterium*, *Bacteroides*, and *Ruminococcus* were more abundant in females. These results are in alignment with a Japanese study that reported higher levels of *Megamonas* in males and higher levels of *Bifidobacterium* in females [79]. A Chinese study also reported a similar pattern, with *Ruminococcus* as the most abundant genera in females [80]. These studies suggest that in terms of microbial composition and the most abundant genera, the Pakistani gut microbiome resembles the microbial profile within the Asian community. Increased abundance of *Proteobacteria* has been linked to gut dysbiosis [81] and obesity [64] and we have also reported an increased abundance of *Proteobacteria* in Pakistani males as compared to females in our previous study [34]. This could be a cause for concern as a positive correlation of *Proteobacteria*, *Prevotella*, and *Bacteroides* was associated with antibiotic resistance genes (ARGs) in a Pakistani cohort [82] and we also observed a similar positive association of these genera including *Proteobacteria (Succinivibrio*, *Cardiobacterium*, *Actinobacillus*, and *Aggregatibacter)* in Pakistani males.

Null modelling analyses further confirmed strong environmental pressure on the gut and oral microbial communities. This is not unexpected given that the host system places an additional constraint on microbial communities [83,84]. We found that dispersal limitation was a key assembly process in gut samples arising from the individual nature of each host, while oral communities showed weak selection and moderate drift (undominated) as the primary assembly process. A complexity stability analysis showed that the oral communities were less stable than the gut communities and thus may have more factors influencing microbial community assembly. Interestingly, we observed gender-based differences whereby female gut communities were influenced by variable selection, and male oral communities were influenced by homogenous selection. Dispersal limitation is reported as a primary assembly process in the human gut microbiome of US and Papa New Guinean populations as well [31]. The authors also noted variable selection as a key process which was influenced by geographical location. However, the authors did not define study participants by gender. In the literature, analyses of the contributions of microbial assembly processes and how different covariates may impact the human microbiome are still relatively rare. Thus, our study provides valuable insight into the impact of the environment (gut, oral, male, and female) and lifestyle (BMI, age, smoking, and antibiotic usage status). The key findings are summarized below in Figure 8.

## 5. Conclusions

Pakistan is categorized as a low-and-middle-income country, and here, the misuse of antibiotics is widespread, and multidrug resistance is prevalent. Our results suggest that in a healthy Pakistani cohort, individuals that consumed tap water had almost 6-fold more associations with piARGs. Therefore, we conclude that drinking water source could be a strong driver in the spread of AMR in Pakistan. To the best of our knowledge, this is one of the first studies of a Pakistani cohort that considers potential AMR profiles of healthy adults with an emphasis on the potential sources of AMR prevalence. The findings highlight the urgent need to improve the quality of drinking water in Pakistan, and the urgent need for antibiotic surveillance of groundwater and tap water.

## Figures and Tables

**Figure 1 microorganisms-11-00279-f001:**
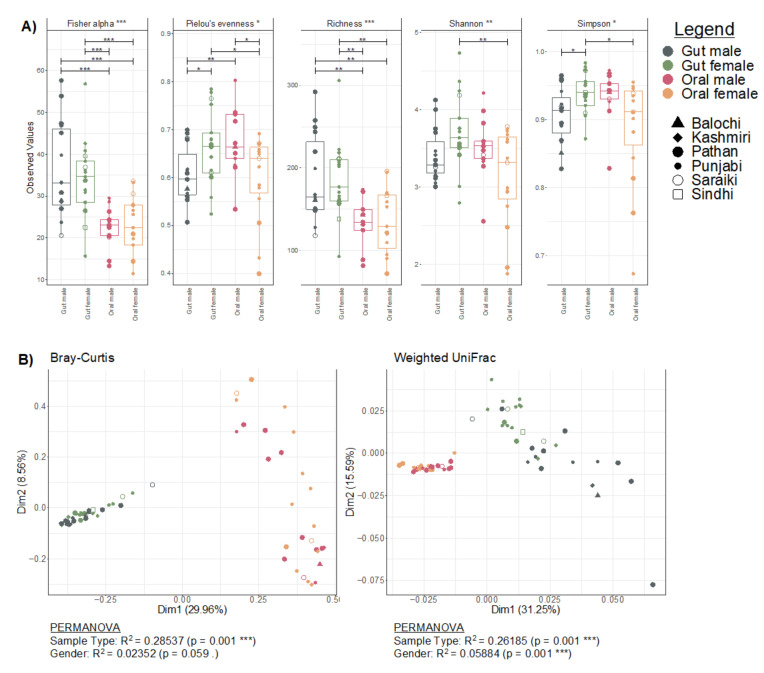
Diversity estimates. (**A**) Commonly used alpha diversity indices with overall significance considering all four cohorts (gut male, gut female, oral male, and oral female) represented in the strip legends whilst pair-wise differences if significant (based on ANOVA) are represented by lines connecting two categories. (**B**) Principal coordinate analysis (PCoA) using different several dissimilarity indices (Bray-Curtis and weighted UniFrac) where ellipses were drawn using 95% confidence intervals based on the standard error of ordination points for a given category. Beneath each figure are the R^2^ values (along with *p*-values if significant) calculated from PERMANOVA. The significances are shown with *p* < 0.1, * *p* <  0.05, ** *p* < 0.01, or *** *p* < 0.001.

**Figure 2 microorganisms-11-00279-f002:**
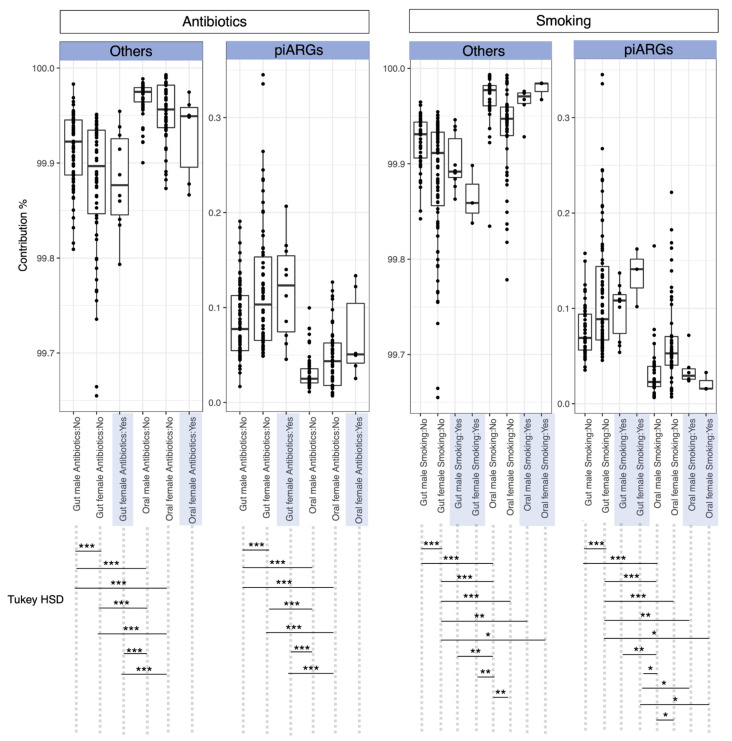
Bray-Curtis contribution of predictive antimicrobial resistance genes (piARGs) to the overall beta diversity within the categories. The plot shows $N\left(N-1\right)/2$ pair-wise differences between samples for each category with left set of panels showing organization by antibiotic status whilst right set of panels show organization by smoking status, respectively. “Others” represent the contribution of the remaining KOs that are not identified as potential AMR genes. Higher contributions represent higher intersample variability in terms of piARGs. Lines connecting categories shows significant relationships (Tukey HSD) with * *p* <  0.05, ** *p* < 0.01, or *** *p* < 0.001.

**Figure 3 microorganisms-11-00279-f003:**
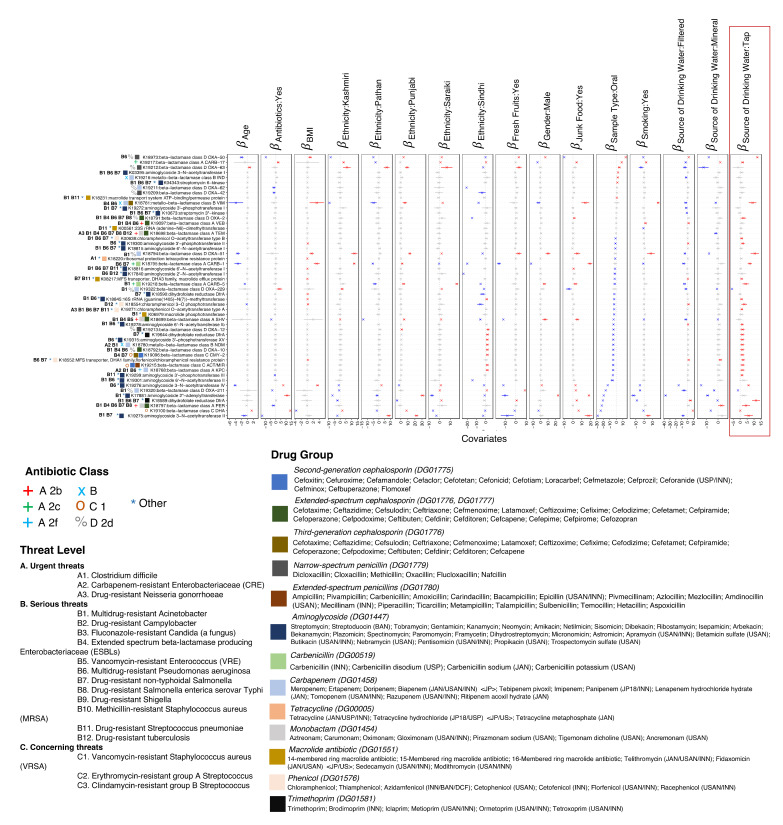
β− coefficients returned from GLLVM procedure for covariates considered in this study by considering the 52 predictive antimicrobial resistance genes (piARGs) detected in this study using PICRUSt2 procedure. Those coefficients which are positively associated with the abundance of a particular piARGs are represented in red colour whilst those that are negatively associated are represented with blue colour, respectively. Where the 95% confidence interval of the β− coefficients cross the 0 boundary, the coefficients are insignificant and are represented by grey colour. Additional annotations comprise of threat levels, classes, and drug groups these piARGs are categorized under.

**Figure 4 microorganisms-11-00279-f004:**
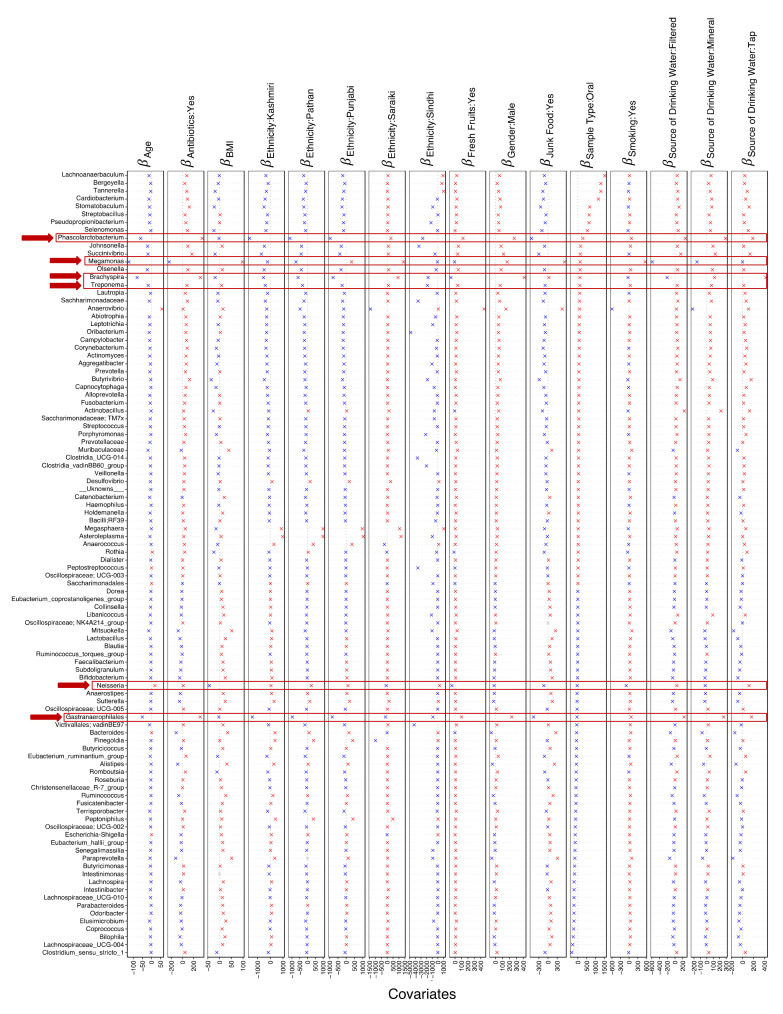
β− coefficients returned from GLLVM procedure for covariates considered in this study. Top 100 most abundant genera were considered, incorporating both continuous data (age, BMI) as well as categorical labelling of samples. Those coefficients which are positively associated with the microbial abundance of a particular species are represented in red colour whilst those that are negatively associated are represented with blue colour, respectively. Since the collation of ASVs was performed at genus level, ASVs that cannot be categorized based on taxonomy are collated under the “__Unknowns__” category. Genera that stand out in terms of significances with multiple covariates are highlighted with a red border.

**Figure 5 microorganisms-11-00279-f005:**
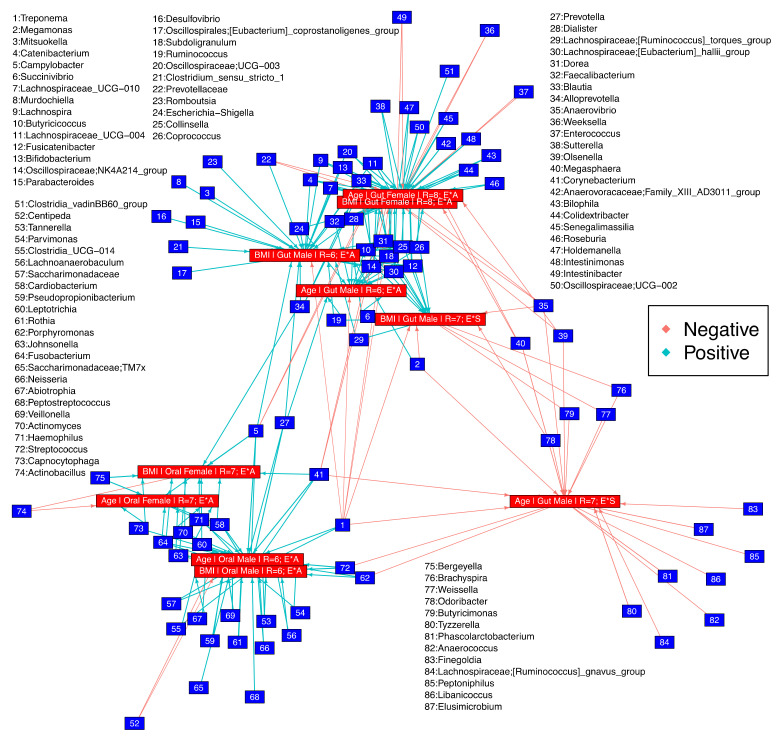
Relationships recovered after applying Hurlbert’s B_N_ to two environmental properties, age and BMI on different sets of environments as a combination of ethnicity and antibiotics usage status (E*A) or ethnicity and smoking status (E*S) with R representing the possible sets of environments. Genera tagged as “Positive” increase in abundance, whilst those tagged as “Negative” decrease in abundance in relationship to the environmental property considered as well as the target set of environments. Further details are given in Appendix A.

**Figure 6 microorganisms-11-00279-f006:**
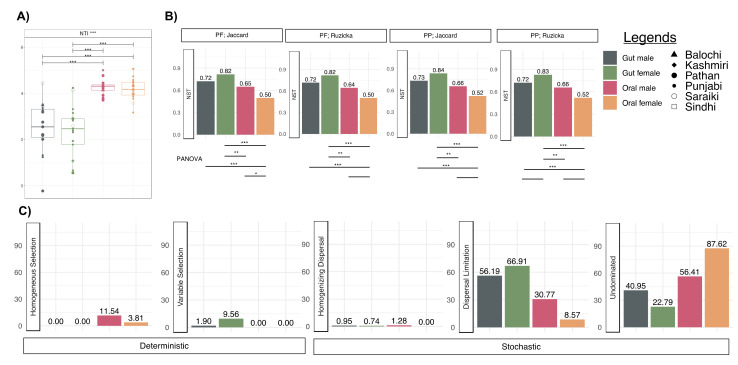
Estimates of microbial community assembly mechanisms structuring samples from four different cohorts (gut male, gut female, oral male, and oral female). (**A**) phylogenetic alpha diversity measure, nearest taxon index (NTI) to elucidate environmental pressures (signficances are calculated with ANOVA). (**B**) Normalised stochasticity ratio (NST) calculated as both incidence based (presence/absence) Jaccard and abundance based Ruzicka metrics with PF and PP being the null modelling regime used. PANOVA connects pairwise categories where the estimates are statistically significant in terms of NST; (**C**) Proportion of assembly processes returned from applying quantitative processing estimate (QPE) for all four cohorts presented as both deterministic and stochastic measures. Lines connecting categories shows significant relationships: blank *p* < 0.1, * *p* < 0.05, ** *p* < 0.01), or *** *p* < 0.001).

**Figure 7 microorganisms-11-00279-f007:**
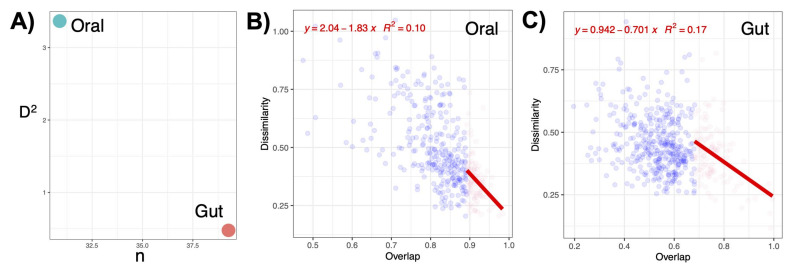
Complexity–stability relationship of oral and gut communities (**A**) where the effective connectance $D^{2}$ is calculated based on fitting a linear regression to top dissimilarity/overlap values for pair-wise samples where the overlap values are in top 25% of communities, and then recovering $D^{2}\;$as the slope of fitted regression for oral (**B**) and gut (**C**) communities.

**Figure 8 microorganisms-11-00279-f008:**
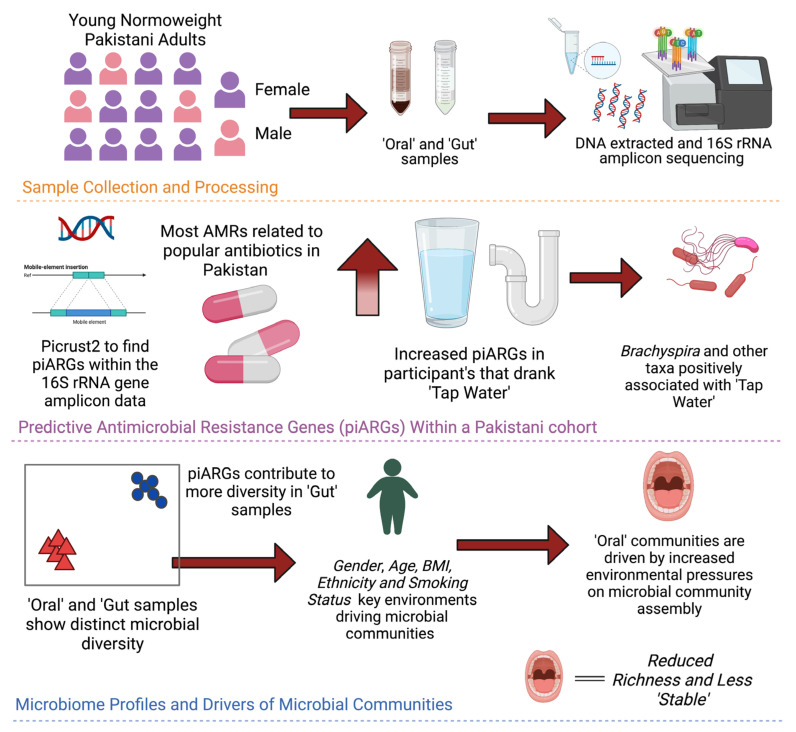
Summary of the key findings of our study. Visualisation was created with BioRender.com.

## Data Availability

Sequencing data and anonymized meta-data are available upon request. Some of the scripts are available as part of statistical packages available at http://userweb.eng.gla.ac.uk/umer.ijaz#bioinformatics (accessed on 1 January 2023).

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
