# Peer review of "A Cross-Sectional Study of Potential Antimicrobial Resistance and Ecology in Gastrointestinal and Oral Microbial Communities of Young Normoweight Pakistani Individuals"

_microorganisms, 2023, doi:10.3390/microorganisms11020279_

Round 1

Reviewer 1 Report (Previous Reviewer 3)

 The manuscript has been significantly improved, although using PICRUSt2 for ARG analysis does not convince me. It is all the better to use qPCR analysis or at least to confirm the prediction using this technique. The genes coding for antibiotic resistance are easily transferred between bacterial species, so associating them with specific taxons is very risky. I think this issue should be discussed in the manuscript. In addition, the authors should state in the abstract that antimicrobial resistance genes were only predicted because it is not clear to the readers.

Author Response

We thank the reviewer for their comments. We understand the reviewer's point re. predictive ARG analysis and have tried to address this within the manuscript. We have now made it clear with modifications throughout the title, abstract, main text and figures (Figure 2 and summary figure) that these are potential AMR genes or predictive ARGs (piARGs). We note that we have used a similar approach to Su et al. (2020) for predictive antimicrobial resistance gene prevalence [piARGs] (who used Tax4Fun which is less reliable than PICRUSt2 (and have included this in the methods section). We have addressed the limitations of the work specifically in the discussion as follows.

“A potential limitation of this work is that the piARG results are based on metabolic profiles predicted through PICRUSt2, which have not been confirmed through quanti-tative PCR or other measures. Moreover, associations with specific taxa need to be con-firmed due to the potential for antibiotic resistance encoding genes to transfer between bacterial species. Nonetheless, PICRUSt2 has been shown to perform well on hu-man-associated microbiome datasets. This is mainly due to a comprehensive reference database of genomes whose functions are already known (a 10-fold increase in the numbers since the previous release) [40]. Shotgun metagenomics of these samples would more accurately highlight ARG prevalence; however, the experimental cost and re-sources for processing and data analysis may be prohibitive for LMICs, and predictive modelling may be a viable monitoring option for determining potential AMR spread.”

Reviewer 2 Report (Previous Reviewer 2)

The authors satisfactorely answered and completed the questions made this Reviewer in the first round of revision and consequently, the manuscript is now adequate for its publication in the current form

Author Response

Thank you for accepting this work for publication and for your time and expertise in the review process. Please note some additional changes have been made in light of newer comments from Reviewer 1.

Reviewer 3 Report (Previous Reviewer 1)

Authors answered all my comments. Therefore, manuscript can be accepted in the present form.

Author Response

Thank you for accepting this work for publication and for your time and expertise in the review process. Please note some additional changes have been made in light of newer comments from Reviewer 1.

Round 2

Reviewer 1 Report (Previous Reviewer 3)

-

This manuscript is a resubmission of an earlier submission. The following is a list of the peer review reports and author responses from that submission.

Round 1

Reviewer 1 Report

The manuscript is interesting. The methodology used is sufficient, and the results support the discussion. However, I have the following comments.

I. Major comments:

1. Improve the writing of the manuscript. It is possible to understand it (English writing), the authors must use a scientific English.

2. In the introduction it is important to briefly include relevant aspects of antimicrobial resistance and ecological aspects, such as excessive use of antibiotics, or the effects of the microbiota on health.

3. In the methodology, the authors must describe important aspects of the study group, such as: number of participants, % of men, women, etc. For this type of study, this information is important. It is not enough to cite a reference (34).

4. The results are interesting, but it was difficult for me to read some figures (size of letters very small). For example, figure 2. Improve the resolution of all figures, please.

5. The discussion is good, but very general. I suggest that the authors include aspects related to health, such as decreased efficacy of antibiotics, antimicrobial resistance and effects on health (respiratory example), use of antibiotics and changes in the microbiota (possible relationship with intestinal pathologies) and briefly discuss possible mechanisms. involved.

II. Minor comments:

1. Improve the wording of the objective of the study.

2. I suggest including a figure (discussion section) that summarizes the main results of the study.

Reviewer 2 Report

The manuscript is wery well written, interesting and the science and method employed are goods. Perhaps the number of samples is a little low and it is not very clear to the reader why the number of "gut" and "oral" samples is different. The authors should clarify a little better the origin of this difference.

There are also some abbreviations that have not been previously described in the text such as "WHO" or "LMICs" (the abstract should be considered a separate section, so it does not count). 

Perhaps, as a major drawback (and this shows that the article has little to criticize), I would say that the title does not reflect well the content of the article, as it mentions "pakistani adults", when the article only includes normopese young people, so I would change "pakistani adults" to young nomoweight pakistani".

Note: At first, I thought the article was plagiarism since it is 75% similar to another one, according to the turnitin software. Then I was able to verify that the similarity was with a preprint of the same article. Given the speed of publication at publishers such as MDPI, I would advise authors not to preprint as this can lead to misunderstandings and cause the article to be unfairly rejected.

Reviewer 3 Report

The authors examined antimicrobial resistance (AMR) and ecology in the gastrointestinal and oral microbial communities of urban Pakistani adults. The authors linked microbial composition and covariates (body mass index, smoking, antibiotic use) to gut microbiome structure and correlated AMR gene prevalence.
The authors showed that tap water was the main cause of AMR development in the Pakistani cohort. In general, the manuscript The results may be of interest to the readership of Microorganisms, but cannot be published in a high-impact journal because it was not clearly written. The major drawback of the manuscript is the lack of information on how antimicrobial resistance genes were detected in the samples studied. The authors used sophisticated statistical tools, but the result is quite limited, because the most valuable results indicate that drinking water is the main source of AMR genes and that smoking can change the composition of the gut microbiome.
The authors should rewrite the manuscript by describing how the AMR genes were analyzed, simplifying the statistical methodology, and highlighting the results of their work in the Conclusions chapter. Ideally, the concentration of the main antibiotics in tap water should be analyzed in the region where the person studied comes from.

Specific comments

1. Abstract: Please rephrase the sentence "Antimicrobial resistance (AMR) is one of the most serious global public health threats.." because it is awkward.
2. Abstract: "We found that drinking water plays a more important role in AMR spread in Pakistan rather than other factors considered" This information was given earlier.
3. Materials and Methods: The analysis of AMR genes should be explained in more detail

4. Results: Figures 3. and 4. should be enlarged

5. Discussion: the first paragraph should be deleted

6. Conclusions: The authors should present here the main results of their work and not what was done.